# Taekwondo Athlete’s Bilateral Achilles Tendon Rupture: A Case Report

**DOI:** 10.3390/medicina59040733

**Published:** 2023-04-08

**Authors:** Jun Young Lee, Sung Hwan Kim, Joo Young Cha, Young Koo Lee

**Affiliations:** 1Department of Orthopaedic Surgery, Chosun University Hospital, 365, Pilmundae-ro, Dong-gu, Gwangju 61453, Republic of Korea; 2Department of Orthopaedic Surgery, Soonchunhyang University Hospital Bucheon, 170, Jomaru-ro, Wonmi-gu, Gyeonggi-do, Bucheon-si 14584, Republic of Korea; shk9528@naver.com (S.H.K.);

**Keywords:** bilateral Achilles tendon rupture, Taekwondo athlete, young athlete

## Abstract

(1) *Background*: Achilles tendon rupture is a common sports injury that may result in severe disability. The overall incidence of Achilles tendon rupture is increasing as a result of growing sports participation. However, cases of spontaneous bilateral Achilles tendon rupture with no underlying disease or risk factors, such as systemic inflammatory disease, steroid or (fluoro)quinolone antibiotics use, are rare. (2) *Objective*: Here, we report a case of a Taekwondo athlete’s bilateral Achilles tendon rupture after kicking and landing. By sharing the experience of treatment and the patient’s course, we suggest one of the possible treatment options and the need to establish a treatment method. (3) *Procedure*: A 23-year-old male Taekwondo athlete visited the hospital, presenting foot plantar flexion failure and severe pain in both tarsal joints, which had occurred upon kicking and landing on both feet earlier that day. During surgery, no degenerative changes or denaturation were observed in the ruptured areas of the Achilles tendons. Bilateral surgery was performed using the modified Bunnel method on the right side and minimum-section suturing on the left side was performed using the Achillon system, followed by lower limb casting. (4) *Result*: Good outcomes were observed on both sides at 19 months postoperatively. (5) *Conclusion*: The possibility of bilateral Achilles tendon rupture during exercise in young subjects with no risk factors should be acknowledged, especially in association with landing. In addition, in athletes, even if there is a possibility of complications, surgical treatment should be considered for functional recovery.

## 1. Introduction

The Achilles tendon (AT) is the largest and thickest tendon in the human body and contributes to foot plantar flexion, hind foot inversion, and even knee flexion [1]. AT rupture is a common sports injury that may result in severe disability. The most common age of AT injury is 30–50 years and it is much more common in men [2]. The overall incidence of AT rupture is increasing as a result of growing sports participation. Spontaneous AT rupture can occur for various reasons and is often reported in patients with systemic inflammatory disease and in those undergoing corticosteroid or (fluoro)quinolone antibiotic treatment or blood dialysis [3]. In patients with peripheral sensory neurological disorders, AT rupture is caused by negative feedback from the physical receptor when the ankle joint is overloaded [4,5,6]. An imbalance between the damage and recovery of the tendon and chronic tendon disease due to degenerative changes can induce AT rupture. Systemic disease and factors like dehydration, secondary hyperparathyroidism, and a reduced blood supply to the tendon in patients on blood dialysis can induce tendon damage via hypoxia and inappropriate metabolism in the tendon [7,8].

AT ruptures are diagnosed by physical examination and imaging studies, which provide additional clinical information [9]. On physical examination, diffuse swelling and bruising are common. When the swelling is severe, a gap may be palpable 2–6 cm proximal to the tendon insertion [10]. Other disease-specific tests are used to confirm the diagnosis, such as the Thompson and Copeland tests [11]. Imaging modalities such as ultrasound and magnetic resonance imaging (MRI) facilitate the diagnosis and monitoring of AT rupture and are used to rule out other injuries [12].

Bilateral spontaneous AT ruptures are uncommon, with an overall incidence of <1% [13]. Most previously reported cases involved risk factors, and there were few papers on the case where there are no risk factors of AT ruptures such as a history of corticosteroid injection or using (fluoro)quinolone [3,14]. To our knowledge, there are no reports in English about bilateral AT rupture in young, healthy Taekwondo athletes with no underlying disease or risk factors. Here, we report the case of a 23-year-old male Taekwondo athlete in whom bilateral AT rupture occurred spontaneously. By sharing our experience of treatment and the patient’s course, we suggest one of the possible treatment options and the need to establish a treatment method.

## 2. Case Presentation

### 2.1. Preoperative Evaluation

A 23-year-old male Taekwondo athlete visited the hospital due to foot plantar flexion failure and severe pain in both tarsal joints, which had occurred upon kicking and landing on both feet earlier that day (Figure 1). The patient had no direct injury, no risk factors for spontaneous AT rupture, and no history of steroid treatment. He had begun to practice Taekwondo when he was in kindergarten and was a university Taekwondo athlete at the time of presentation. He reported no previous experience of pain in the bilateral ATs for 16 years from the start of exercise.

At the time of admission, the Thompson squeeze test was positive in both ankles, and digital exploration revealed dimpling 6 cm superior to the calcaneal tendon attachment site. Preoperative ankle radiographs showed no abnormality in the bone or surrounding soft tissue, other than the loss of Kager’s triangle (Figure 2). MRI confirmed the diagnosis of bilateral AT rupture (Figure 3).

### 2.2. Surgical Procedure

Surgery was performed with the patient under general anesthesia and in the prone position. We decided to do surgery on the right side first. The ruptured ATs were exposed via posteromedial skin incisions, and invasive tendinosuture was performed (Figure 4). After incising the paratenons, the ATs were sutured securely using the modified Bunnel method. The peritenon tissues surrounding the ATs were also sutured sufficiently to maintain blood circulation to the greatest degree possible. During surgery, complete oblique rupture of the ATs was observed approximately 6 cm superior to the calcaneal tendon attachment sites, with no additional findings. On the left side, the area of AT rupture was explored digitally, and an approximately 3 cm skin incision was made (Figure 5a). Minimally invasive suturing was performed by passing an ETHIBOND no. 2 suture in parallel from the medial to the lateral side using the Achillon tendon suture system (The Achillon Technique Guide, No. 2 FiberWire; Arthrex, Naples, FL, USA), and then crossing it over to pass approximately 1 cm distal to the ruptured area. The same procedure was performed proximal to the ruptured area, and the suture was passed to the distal site after plantar flexion of the ankle to enable passage into the interior, followed by strain knotting (Figure 5b). Then, the left side, including the peritenon, was sutured anatomically, followed by suturing of the surgical wound. Casts were applied to both feet in complete plantar flexion, and the wounds were sterilized by opening windows in the casts.

### 2.3. Postoperative Care

The patient’s hospitalization was uneventful. The sutures on both sides were removed, short lower limb casts were applied 2 weeks postoperatively, and the patient was discharged. The patient underwent a 6-week course of physical therapy with assisted 90° movement of the ankle joint. At 3 months after the operation, clinical evaluation was done using the American Orthopedic Foot and Ankle Society (AOFAS) ankle-hindfoot functional score, the Achilles Tendon total Rupture Score (ATRS), and Visual Analog Scale (VAS) scores. His AOFAS ankle-hindfoot functional score, ATRS, and VAS scores were 85, 77, and 3 points, respectively. He began jogging and one-heel squatting at 3 months postoperatively and resumed Taekwondo practice approximately 8 months postoperatively. He had regained normal range of motion at 19 months after the operation (Figure 6). At the final follow-up, the scoring methods performed at 3 months were investigated once more. The AOFAS ankle-hindfoot functional score, ATRS, and VAS were 100, 97, and 1, respectively. Other complications such as infection, rerupture, and nerve damage did not occur and MRI of the ankle joints, which was performed 19 months after the operation, confirmed that both ATs were well maintained (Figure 7).

## 3. Discussion

Our patient was a healthy 23-year-old male with no previous symptoms or lesions of the AT; thus, general degenerative changes were unlikely to be present. He was very healthy with no underlying disease and had no risk factors for AT rupture, such as a history of corticosteroid injection or using (fluoro)quinolone, as mentioned above. This bilateral AT rupture was likely caused by a high-energy bilateral load upon landing. We study and report a fairly rare case, and it is hard to find articles that treated both sides in different ways. We suggest one of the possible treatment options and the need to establish a treatment method in cases similar to this one.

Taekwondo is the national martial art of the Republic of Korea and an official Olympic sport. It involves a combination of movements, with “Poomsae”, emphasizing mental concentration and defense-orientated movements using the whole body, and “Gyorugi”, being more offensively orientated; the fists and feet are used to hit and stomp, respectively [15]. The sport is normally done bare-footed. As a result, Taekwondo athletes tend to have injuries involving the lower extremities and ankles [16,17]. As with many other martial arts where little to no protective gear is allowed on the legs and ankles, the forces acting on the foot and ankle can be great [18]. Excessive loading can cause the rupture of a healthy AT [5]; this can occur during abrupt plantar flexion. Habusta reported bilateral AT rupture in a gymnast caused by simultaneously raising both feet from a lowered position [19].

The treatment options for AT rupture include various invasive and percutaneous surgical procedures and conservative treatment. Surgical treatment has more complications but reduces the likelihood of rerupture [20]. Since our patient was an athlete, it was important to reduce the possibility of rerupture, have early motion, and have rehabilitation. Thus, even though there was a possibility of complications, surgical treatment was selected on the bilateral side at the same time. According to some studies, complications associated with invasive suturing can be reduced by percutaneous surgery [8,21,22]. In this case, as the patient is right-footed, he requested definite repair under direct vision using open technique on the right side. So, invasive suturing was performed on the right side and minimal suturing was performed on the left side. The patient recovered a normal range of movement in both ankles, and there were no wound-related problems. This case demonstrates that the clinical outcomes of invasive tendon suturing and minimum section suturing to treat the same injury do not differ markedly.

This study has several limitations. First, there was no follow-up after the patient returned to Taekwondo matches. Therefore, return to sport could not be evaluated. Second, no specific rehabilitation protocol was applied; unlike unilateral AT rupture, there is no specific rehabilitation protocol for bilateral AT rupture because the frequency is low. Since bilateral AT rupture is expected to increase with the frequency of AT rupture, it is necessary to establish a specific protocol by studying many cases through a multi-institutional study.

## 4. Conclusions

In summary, the possibility of bilateral AT rupture during exercise in young subjects with no risk factors should be acknowledged, especially in association with landing. In addition, in athletes, even if there is a possibility of complications, surgical treatment should be considered for functional recovery.

## Figures and Tables

**Figure 1 medicina-59-00733-f001:**
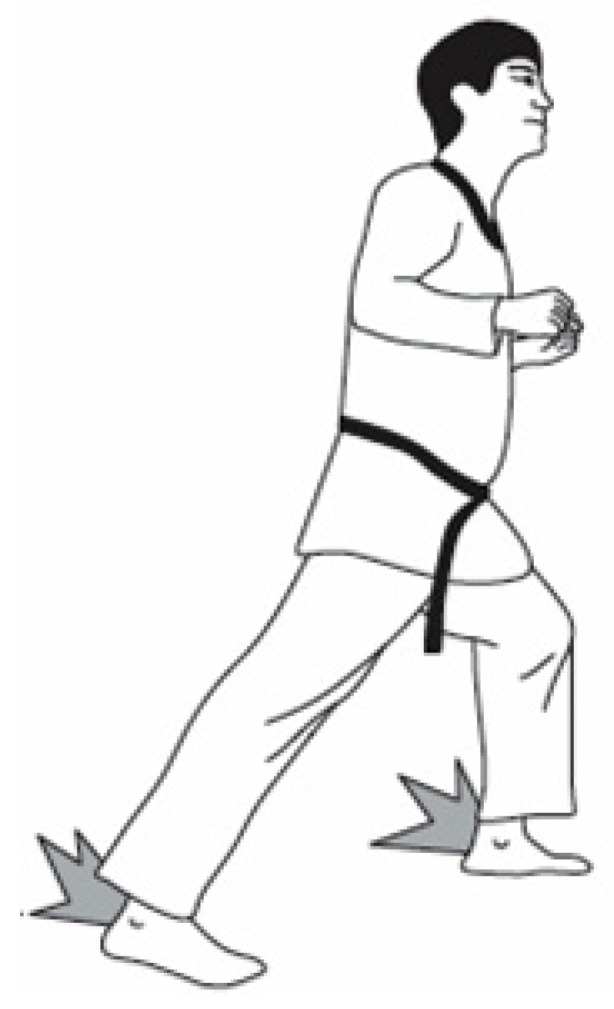
Schematic illustration of the mechanism of injury in a young Taekwondo athlete after kicking and landing on both feet simultaneously.

**Figure 2 medicina-59-00733-f002:**
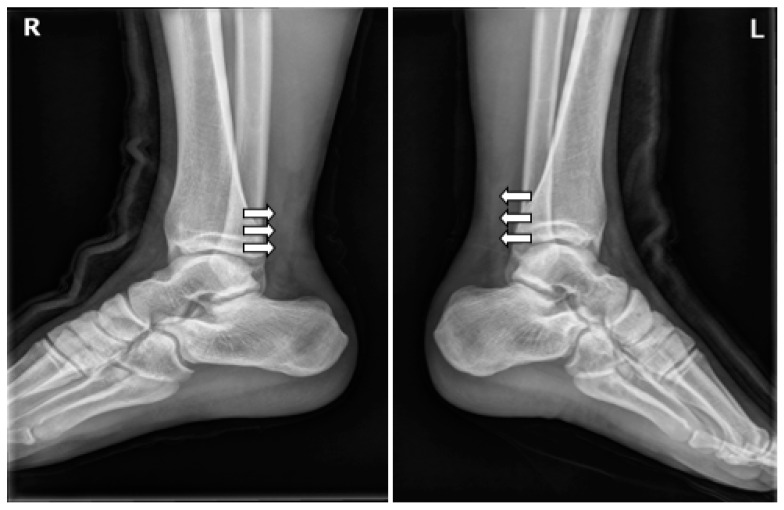
Lateral conventional radiographs of both ankles following bilateral Achilles tendon rupture show marked thickening of the Achilles tendons, loss of the normal sharp anterior borders (white arrows), and effacement of the pre-Achilles/Kager’s fat pad.

**Figure 3 medicina-59-00733-f003:**
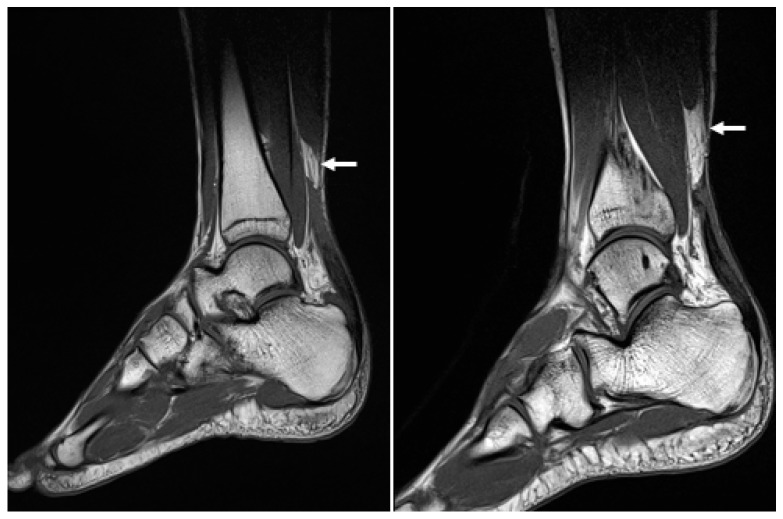
Preoperative sagittal magnetic resonance images of both ankles show rupture of the Achilles tendons (white arrow).

**Figure 4 medicina-59-00733-f004:**
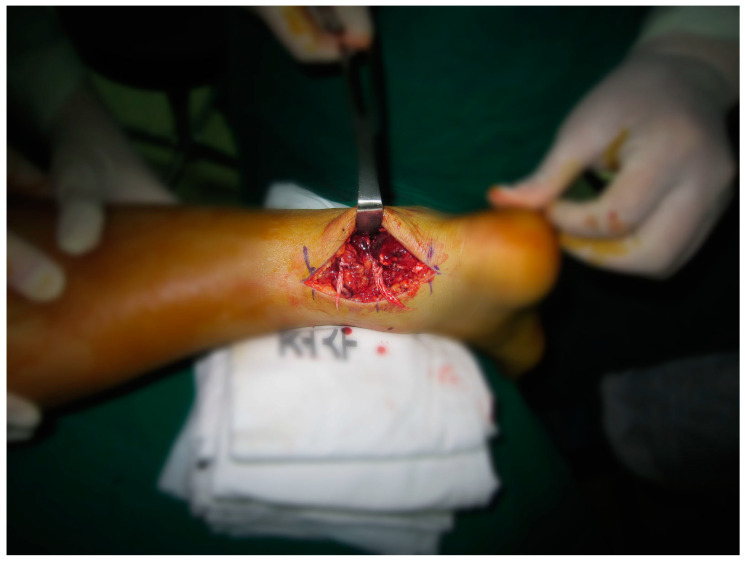
Intraoperative finding of Achilles tendon rupture in the right ankle.

**Figure 5 medicina-59-00733-f005:**
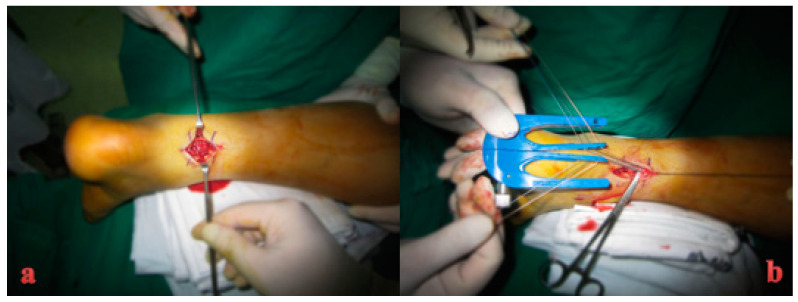
(**a**) Intraoperative finding of Achilles tendon rupture in the left ankle. (**b**) The Achillon^®^ instrument was introduced proximally under the paratenon, and three needles were passed.

**Figure 6 medicina-59-00733-f006:**
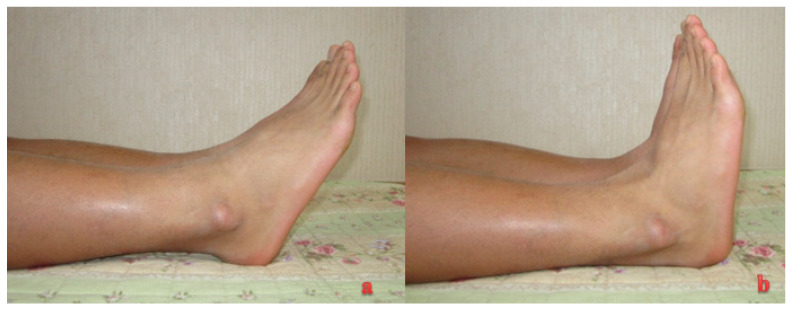
Postoperative clinical outcome (19 months after operation): (**a**) plantar flexion and (**b**) dorsiflexion.

**Figure 7 medicina-59-00733-f007:**
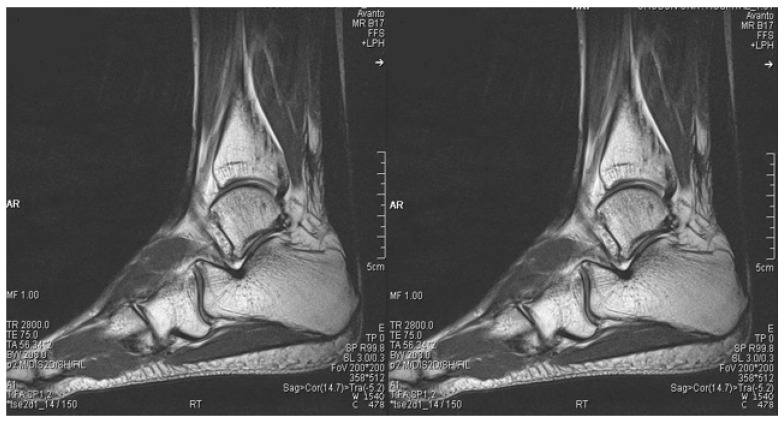
Postoperative (19 months after operation) sagittal magnetic resonance images of both ankles showing good continuity of the Achilles tendons.

## Data Availability

Data sharing is not applicable to this article because any datasets were made or analyzed during this study.

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
