# Peer review of "Taekwondo Athlete’s Bilateral Achilles Tendon Rupture: A Case Report"

_medicina, 2023, doi:10.3390/medicina59040733_

Round 1

Reviewer 1 Report

Dear Authors,

Thanks!

Abstract:

Please, insert: objetive, procedures, results, conclusions and practical implications

Introduction:

In scientific writing, the state of the art describes the current knowledge about the studied matter through the analysis of similar or related published work. So, please, we need an explanation about: “Taekwondo athlete’s bilateral achilles tendon rupture” (please, insert recent references/studies_2022_2023).

Please, we need an explanation about: the dependent (cause) and independent (effect) variables and the hypotesis of the study. In this sense, hypothesis states your predictions about what your research will find.

+

(…) “Bilateral spontaneous AT ruptures are uncommon, with an overall incidence of < 1% 52 [13]. Most previously reported cases involved risk factors. To our knowledge, there are no 53 reports in English about bilateral AT rupture in young, healthy Taekwondo athletes with 54 no underlying disease or risk factors, such as steroid use. Here, we report the case of a 23- 55 year-old male Taekwondo athlete in whom bilateral AT rupture occurred spontaneously.” (…)

Please, insert research objective(s), hypothesis “testing” and pinpoint the major focus of your research.

Case Presentation

2.1. Preoperative evaluation

Sample/participants

(…) A 23-year old male Taekwondo athlete… (----------insert mean, SD, CV)

Sample: Inclusion and exclusion criteria? Please, explain.

“He had begun to practice Taekwondo when he was in kindergarten, and was a university Taekwondo athlete at the time of presentation. He reported no previous experience of pain in the bilateral ATs.”------------ ( years of practice experience?????).

Figure 1. Left Schematic illustration of the mechanism of injury in a young Taekwondo athlete  after kicking and landing on both feet simultaneously (adptated from????)

“At the time of admission, the Thompson squeeze test was positive in both ankles and 65 digital exploration revealed dimpling 6 cm superior to the calcaneal tendon attachment 66 site. Preoperative ankle radiographs showed no abnormality in the bone or surrounding 67 soft tissue, other than the loss of Kager’s triangle (Fig. 2). MRI confirmed the diagnosis of 68 bilateral AT rupture (Fig. 3).”

Please, selecting and justifying your Research Design with studies.

“The same procedure was performed proximal to the ruptured area, and the suture was 99 passed to the distal site after plantar flexion of the ankle to enable passage into the interior, 100 followed by strain knotting (Fig. 5b). Then, the left side, including the peritenon, was su- 101 tured anatomically, followed by suturing of the surgical wound. Casts were applied to 102 both feet in complete plantar flexion, and the wounds were sterilized by opening windows 103 in the casts.”

Please, selecting and justifying your procedures with studies.

2.3 Postoperative care

Please, selecting and justifying your procedures with studies.

Discussion

Line 147-158 “ He was very healthy and had no risk factors for AT rupture”.

Is there a relationship (i.e., studies) between “health” and “risk factos” for AT rupture?

 +

Please:

Summarize your results and outline their interpretation in light of the published literature

Explain the importance of your results

Acknowledge the shortcomings of the study

Discuss any future directions

 Pick your tenses carefully

5. Conclusions

 Is not merely a summary of the main topics covered or a restatement of your research problem….

References

2022= 1 Study

2023= 0 Studies

Total=21

Thanks

Kind regards

Author Response

Dear review,

We really appreciate to your kind review comments.

Actually, those comments are really helpful to our article to be more completed.

Below are our answer to your comments and questions.

Abstract:

Please, insert: objetive, procedures, results, conclusions and practical implications

-> We insert background, objetive, procedures, results, conclusions. Actually, practical implications have the same content as conclusions and tends to overlap, so it was combined.

Introduction:

In scientific writing, the state of the art describes the current knowledge about the studied matter through the analysis of similar or related published work. So, please, we need an explanation about: “Taekwondo athlete’s bilateral achilles tendon rupture” (please, insert recent references/studies_2022_2023).

 -> According to your advice, we tried to find recent articles. However, we couldn’t find article about “Taekwondo athlete’s bilateral achilles tendon rupture” in english which is published recently. Instead, papers on similar topics were added as references. (Line 59-61)

Please, we need an explanation about: the dependent (cause) and independent (effect) variables and the hypotesis of the study. In this sense, hypothesis states your predictions about what your research will find.

+ 

(…) “Bilateral spontaneous AT ruptures are uncommon, with an overall incidence of < 1% 52 [13]. Most previously reported cases involved risk factors. To our knowledge, there are no 53 reports in English about bilateral AT rupture in young, healthy Taekwondo athletes with 54 no underlying disease or risk factors, such as steroid use. Here, we report the case of a 23- 55 year-old male Taekwondo athlete in whom bilateral AT rupture occurred spontaneously.” (…)

Please, insert research objective(s), hypothesis “testing” and pinpoint the major focus of your research.

 -> Actually, the limitation of our report is that there is only one case. So, it was hard to set the dependent (cause) and independent (effect) variables. Instead, we added objective of our reports as hypothesis (Line 63-66). If we collect several cases later and write original, maybe we can use the treatment method as a hypothesis at that time. At that time, we will not forget your advice and reflect it.

Case Presentation

2.1. Preoperative evaluation

Sample/participants

(…) A 23-year old male Taekwondo athlete… (----------insert mean, SD, CV)

Sample: Inclusion and exclusion criteria? Please, explain.

 -> Actually, the limitation of our report is that there is only one case. We don’t have any other cases it is hard to set inclusion and exclusion criteria. It is same reason that there is no value of mean, SD, CV.

“He had begun to practice Taekwondo when he was in kindergarten, and was a university Taekwondo athlete at the time of presentation. He reported no previous experience of pain in the bilateral ATs.”------------ ( years of practice experience?????).

-> We added the duration of practice experience (Line 74-75) “He reported no previous experience of pain in the bilateral ATs for 16 years from the start of exercise until he felt pain in adult”

Figure 1. Left Schematic illustration of the mechanism of injury in a young Taekwondo athlete  after kicking and landing on both feet simultaneously (adptated from????)

-> We understood your question as asking where the source was. The photo was made by ourselves. We might be sorry if we answered incorrectly.

“At the time of admission, the Thompson squeeze test was positive in both ankles and 65 digital exploration revealed dimpling 6 cm superior to the calcaneal tendon attachment 66 site. Preoperative ankle radiographs showed no abnormality in the bone or surrounding 67 soft tissue, other than the loss of Kager’s triangle (Fig. 2). MRI confirmed the diagnosis of 68 bilateral AT rupture (Fig. 3).”

Please, selecting and justifying your Research Design with studies.

-> If what you're talking about is study design, then its level of evidence is 5, case report. If you have a different meaning, please explain in detail and we will work hard to supplement it. 

“The same procedure was performed proximal to the ruptured area, and the suture was 99 passed to the distal site after plantar flexion of the ankle to enable passage into the interior, 100 followed by strain knotting (Fig. 5b). Then, the left side, including the peritenon, was su- 101 tured anatomically, followed by suturing of the surgical wound. Casts were applied to 102 both feet in complete plantar flexion, and the wounds were sterilized by opening windows 103 in the casts.”

Please, selecting and justifying your procedures with studies.

-> If what you're talking about is study design, then its level of evidence is 5, case report. If you have a different meaning, please explain in detail and we will work hard to supplement it (same as above question)

2.3 Postoperative care

Please, selecting and justifying your procedures with studies.

-> If what you're talking about is study design, then its level of evidence is 5, case report. If you have a different meaning, please explain in detail and we will work hard to supplement it (same as above question)

Discussion

Line 147-158 “ He was very healthy and had no risk factors for AT rupture”.

Is there a relationship (i.e., studies) between “health” and “risk factos” for AT rupture?

 -> We used words “healthy” as no underlying disease. And “risk factors” means well known risk factors of achilles tendon rupture (for example, history of corticosteroid injection or using (fluoro)quinolone). To avoid ambiguity, we fixed sentences to be read easily. (Line 150-152) “He was very healthy with no underling disease and had no risk factors for AT rupture like history of corticosteroid injection or using (fluoro)quinolone, as mentioned above.”

Please:

Summarize your results and outline their interpretation in light of the published literature

-> We summarize our results and interpretation in light of the published literature. (Line 153-155) “We study and report a fairly rare case, and it is hard to find articles that treated both side in different ways.”

Explain the importance of your results

-> We explain the importance of our results. (Line 155-156) “We suggest one of the possible treatment option and the need to establish a treatment method in cases similar to this one.”

Acknowledge the shortcomings of the study

-> We write our study’s limitation in Line 177-180

Discuss any future directions

-> We write future directions in Line 180-183

 Pick your tenses carefully

-> We fill very sorry for some unsuited tenses and fixed words.

  1. Conclusions

Is not merely a summary of the main topics covered or a restatement of your research problem….

-> Because this study covered only one case, it is too early to draw broader conclusions. So, in this study, we described only our observations in conclusion. Later, when we collect more cases and write original article, we will reflect your advice and write a better article.

Reviewer 2 Report

 Taekwondo athlete’s bilateral achilles tendon rupture : A case  report and literature review

Review

L16 - “ataxia” is a central cause – use an easier word or phrase

Abstract – avoid using abbreviations

L37 – Corticosteroid?? Use the correct word

L40 – remove ‘to”

General – when referring to references and there is more than 1, either use a hyphen or comma – be consistent

L55 – remove “such as steroid use”

L127 – “consider changing “did not exist” to “did not occur”

L130 – figure 6 – mention at what timeframe this was

L133 – figure 3 – mention the time frame

L136 – fully write out “Korea”’s name as known internationally

L137 – change “motions” to “movements”

L139-140 – Consider changing to – “the sport is done bare-footed”

L149 – consider changing to ‘high energy load bilaterally upon landing”

L153-154 – was this the only reason? Why both in the same setting? Why not delayed surgery for one? It will be good to mention more details on the choice

L155-160 – perhaps explain why different approaches were done for the different sides

L161-162 – but you mention they starting returning at 3 months, then 8 months then full range by 19 months – perhaps be clearer about this as a limitation

L173-174 – remove this “For research articles with several authors, a short paragraph specifying their 173 individual contributions must be provided. The following statements should be used”

The ethic approval document must be included

Review your introduction – very much in the similar lines as your Reference number 3

Author Response

Dear review,

We really appreciate to your kind review comments.

Actually, those comments are really helpful to our article to be more completed.

Below are our answer to your comments and questions.

L16 - “ataxia” is a central cause – use an easier word or phrase

-> We change ataxia to inability to foot plantar flexion

A 23-year old male Taekwondo athlete visited the hospital presenting inability to foot plantar flexion and severe pain in both tarsal joints which had occurred upon kicking and landing on both feet earlier that day.

Abstract – avoid using abbreviations

-> We change AT to achilles tendon

L37 – Corticosteroid?? Use the correct word

-> We changed steorid to corticosteroid

L40 – remove ‘to”

-> We removed ‘to’ from sentence.

General – when referring to references and there is more than 1, either use a hyphen or comma – be consistent

-> Thanks for your advice. We tried to maintain consistency but we need to follow medicina’s reference format. For example, if there are two references, applying comma is medicina’s format and using a hyphen in case of three or more references.

L55 – remove “such as steroid use”

-> We removed “such as steroid use”

L127 – “consider changing “did not exist” to “did not occur”

-> We changed “did not exist” to “did not occur”

L130 – figure 6 – mention at what timeframe this was

-> We added (19 months after operation)

L133 – figure 3 – mention the time frame

-> We added (19 months after operation)

L136 – fully write out “Korea”’s name as known internationally

-> We changed korea to republic of korea

L137 – change “motions” to “movements”

-> We changed motions to movements

L139-140 – Consider changing to – “the sport is done bare-footed”

-> We changed phrases to The sport is done normally bare-footed.

L149 – consider changing to ‘high energy load bilaterally upon landing”

-> We changed to high energy load bilaterally upon landing

L153-154 – was this the only reason? Why both in the same setting? Why not delayed surgery for one? It will be good to mention more details on the choice

-> At first, to decrease wound size, we planned to having surgery with MIS technique on both foot.

However, as patient is athelete, he wanted to have early motion and rehabilization, decrease re-rupture rate. So we decided to treat patient by surgery on both foot at same time.

L155-160 – perhaps explain why different approaches were done for the different sides

-> As patient is right footed, patient requested definite repair under direct vision using open technique on right side and MIS technique on left side for minimizing wound size

L161-162 – but you mention they starting returning at 3 months, then 8 months then full range by 19 months – perhaps be clearer about this as a limitation

-> The “exercise” means acutal taekwondo matches. As this word could draw misunderstand, we changed exercise to taekwondo matches.

L173-174 – remove this “For research articles with several authors, a short paragraph specifying their 173 individual contributions must be provided. The following statements should be used”

-> We removed this sentences. Thanks for your advice and we feel sorry about this kind of mistake

Round 2

Reviewer 1 Report

Dear Authors,

Thanks!

Nice job!

Kind regards

Author Response

Dear reviewer,

Thanks for your kind comments

King regards